# Glia Cells Control Olfactory Neurogenesis by Fine-Tuning CXCL12

**DOI:** 10.3390/cells12172164

**Published:** 2023-08-28

**Authors:** André Dietz, Katja Senf, Julia Karius, Ralf Stumm, Eva Maria Neuhaus

**Affiliations:** Pharmacology and Toxicology, Jena University Hospital, Friedrich Schiller University Jena, Drackendorfer Str. 1, 07747 Jena, Germany; andre.dietz@med.uni-jena.de (A.D.); katja.senf@med.uni-jena.de (K.S.); julia.karius@med.uni-jena.de (J.K.); ralf.stumm@med.uni-jena.de (R.S.)

**Keywords:** olfactory, stem cell, CXCR4, ACKR3, CXCR7, CXCL12, scavenging, heparan sulfate, neurogenesis

## Abstract

Olfaction depends on lifelong production of sensory neurons from CXCR4 expressing neurogenic stem cells. Signaling by CXCR4 depends on the concentration of CXCL12, CXCR4’s principal ligand. Here, we use several genetic models to investigate how regulation of CXCL12 in the olfactory stem cell niche adjusts neurogenesis. We identify subepithelial tissue and sustentacular cells, the olfactory glia, as main CXCL12 sources. Lamina propria-derived CXCL12 accumulates on quiescent gliogenic stem cells via heparan sulfate. Additionally, CXCL12 is secreted within the olfactory epithelium by sustentacular cells. Both sustentacular-cell-derived and lamina propria-derived CXCL12 are required for CXCR4 activation. ACKR3, a high-affinity CXCL12 scavenger, is expressed by mature glial cells and titrates CXCL12. The accurate adjustment of CXCL12 by ACKR3 is critical for CXCR4-dependent proliferation of neuronal stem cells and for proper lineage progression. Overall, these findings establish precise regulation of CXCL12 by glia cells as a prerequisite for CXCR4-dependent neurogenesis and identify ACKR3 as a scavenger influencing tissue homeostasis beyond embryonic development.

## 1. Introduction

Stem cell niches provide a highly specialized micromilieu made up of the extracellular matrix, cell–cell contacts, growth factors, and other small molecules to ensure maintenance, self-renewal, and differentiation of stem cells [1]. Astrocytes are a crucial part of all adult neurogenic niches, and they play an important role in instructing and regulating adult neural stem cells. As well as glial cells, the extracellular matrix is equally important for the development of new neurons from stem cells [2].

The stem cell niche in the olfactory epithelium (OE) generates new neurons throughout life to regenerate continuously dying olfactory neurons [3,4,5]. It is composed of glia cells, called sustentacular cells, and horizontal and globose basal stem cells. Globose basal cells (GBCs) represent the actively proliferating neurogenic stem cells of the OE which are essential to maintain the olfactory sense [6,7], but little is known about microenvironmental cues regulating their proliferation and differentiation [8,9]. Horizontal basal cells (HBCs) are quiescent reserve stem cells [10] that renew sustentacular cells [11]. Apart from that, HBCs amplify inflammatory signaling via NF-κB, and thereby contribute to maintaining local immune defense [12]. Supporting cells are radially oriented cells that traverse the OE from the laminar propria to the apical surface and form structures that envelop the neuronal cells. These cells provide structural support to olfactory neurons, act as phagocytes, and are involved in the metabolism of external compounds. Moreover, the cells release molecules involved in neuroprotection such as Neuropeptide Y [13], endocannabinoids [14], and insulin [15].

We recently identified C-X-C chemokine receptor 4 (CXCR4) as an essential regulator of olfactory neurogenesis [16]. Functional CXCR4 signaling in tissues depends critically on the level of its ligand, C-X-C motif chemokine ligand 12 CXCL12 [17]. CXCL12 is an important paracrine mediator in a variety of stem cell niches, but little is known about how CXCL12 is presented to stem cells and how its availability in niches is regulated. Given the relevance of CXCL12 in regenerative, neoplastic, and inflammatory settings, a better understanding of how CXCL12 is presented to stem cells may provide avenues to new therapeutic strategies [18,19,20]. Local CXCL12 levels can be adjusted by atypical chemokine receptor 3 (ACKR3)/formerly CXCR7. ACKR3 binds CXCL12 with high affinity [21,22] and, after internalization of the ACKR3/CXCL12 complex, enables its intracellular degradation [21,22]. This mechanism depends on continuous cycling of ACKR3 between the cell membrane and the cytoplasm [23,24], a process tightly regulated by CXCL12-induced ACKR3 phosphorylation [17,25]. Notably, ACKR3-mediated CXCL12 scavenging occurs in cells undergoing active CXCL12/CXCR4 signaling and in CXCR4-negative cells. Most chemokines, including CXCL12, are regulated by scavenging receptors as well as immobilized glycosaminoglycans (GAG). It is necessary for the correct presentation and gradient formation of chemokines in tissues and the circulation [26,27], e.g., presenting CXCL12 towards hematopoietic progenitor cells in the bone marrow [28]. Despite their dispensability for proper development, CXCL12/GAG interactions play a critical role in tissue regeneration, particularly in vascular growth after injury [29]. Conversely, ACKR3-mediated scavenging has only been documented in mammals [30,31,32,33] and zebrafish [17,34,35] during development. Currently, no research has examined how scavenging and GAG binding work together to regulate CXCL12 function.

Here, we describe how sustentacular cells regulate CXCR4-dependent olfactory neurogenesis by scavenging CXCL12 and by controlling CXCL12 expression. While CXCL12 binding to the extracellular matrix in HBCs is required for GBCs to get access to CXCL12, glial ACKR3 scavenges CXCL12 and thereby regulates CXCR4 signaling in a non-cell autonomous manner. Notably, ACKR3 and CXCR4 are strictly non-overlapping in the OE, excluding the possibility that ACKR3 modulates CXCL12 signaling via heterodimerization with CXCR4. The collective interaction of neuronal stem cells with glial cells and the extracellular matrix is required for the generation of newly formed olfactory neurons during adult neurogenesis.

## 2. Materials and Methods

### 2.1. Animal Breeding and Treatment

Animal experiments were conducted in accordance with the EC directive 86/609/European Economic Community guidelines for animal experiments and were permitted by the local government (Thüringer Landesamt für Lebensmittelsicherheit und Verbraucherschutz). Mice were kept under 12 h light/dark cycles with *ad libitum* access to food and water. C57BL6/6J wild-type mice were originally purchased from Charles River Laboratories (Sulzfeld, Germany) and sacrificed at P8, 6 weeks, and 8 weeks. Tg(*Krt14-cre*)1Amc mice were purchased from Charles River Laboratories (Sulzfeld, Germany, #004782 [36]). In these mice, the transgene is composed of a Cre recombinase gene under the control of a Keratin 14 (*Krt14*) promoter. *Krt14* is expressed in HBCs. Tg(*Mpz-Cre*)26Mes mice were purchased from Charles River Laboratories (Sulzfeld, Germany); the Cre recombinase gene is under the control of a mouse Mpz (myelin protein zero) promoter [37], and MPZ is expressed in sustentacular cells. Expression of tdT was used to control for the expected localization of promoter activation. ACKR3^LoxP/LoxP^ [38] and CXCL12^LoxP/LoxP^ mice [39] were on a C57BL6/J background. Transgenic CXCL12-RFP mice, expressing CXCL12-RFP under *Cxcl12* promoter, were generated by Prof. R. J. Miller [40] and bred on a C57BL6/J background. Monomeric red fluorescence protein 1 (mRFP1) was inserted at the end of the CXCL12 coding sequence to generate a CXCL12-mRFP1 fusion construct in a CXCL12-containing BAC clone (RP23–203H21) [40]. CXCL12-RFP-transgenic mice contain two to five additional copies of functional *Cxcl12* and have been used before as a model for *Cxcl12* overexpression [25]. HA-ACRK3-ST/A and HA-ACKR3 mouse models were described recently [25]; an HA-tag was inserted in-frame with the ATG in exon 2 in order to produce an HA-ACKR3 fusion protein. For the HA-ACKR3-ST/A mutant strain, the following mutations were inserted into the wild-type exon 2 sequence: S335A, S347A, S350A, S355A, T338A, T341A, T352A, and T361A. Transgenic *Ackr3*-GFP mice (bred on a C57BL6/J background) were received from the Gene Expression Nervous System Atlas project GENSAT; eGFP is expressed in the cells when the CXCR7 promoter is active. The following mice were generated in this study: Tg(*Mpz-cre*); *Ackr3*^LoxP/LoxP^; R26^CAG-LSL-tdT^ mice in order to delete *Ackr3* in *Mpz*-expressing sustentacular cells. Tg(*Krt14-cre*); *Cxcl12*^LoxP/LoxP^; R26^CAG-LSL-tdT^ mice in order to delete *Cxcl12* in *Krt14*-expressing HBCs. Tg(*Mpz-cre*); *Cxcl12*^LoxP/LoxP^; R26^CAG-LSL-tdT^ mice in order to delete *Cxcl12* in *Mpz*-expressing sustentacular cells. All genotypes generated are summarized in Appendix A. For regeneration experiments, mice at an age of 8 weeks were treated once with 50 mg/kg methimazole (Sigma-Aldrich, St. Louis, MO, USA) or 0.9% sodium chloride as a control intraperitoneally and euthanized with an overdose of isoflurane at 3, 14, or 28 days post-injection (dpi).

### 2.2. Immunofluorescence and Tissue Preparation

For all experiments, mice were directly decapitated (young mice, P8) or decapitated after euthanasia with isoflurane (adult mice, 6–8 weeks). Whole heads (P8) or extracted OEs (6–8 weeks, according to a modified protocol [41]) were fixed in 4% paraformaldehyde (PFA; Carl Roth, Karlsruhe, Germany) for 24 h at 4 °C, cryopreserved in 30% sucrose and frozen in 2-Methylbutane (Carl Roth, Karlsruhe, Germany) for immunofluorescence or directly frozen in 2-Methylbutane for in situ hybridization. Tissues were fixed in tissue-freezing medium (Leica Microsystems, Wetzlar, Germany) on a specimen disk and coronary sectioned at 18 µm thickness. The sections were immunostained as previously described [16]. Used primary and secondary antibodies are listed in Appendix A; nuclear staining was performed with Hoechst (Thermo Fisher Scientific, Waltham, MA, USA)**.** Images were obtained using a confocal laser scanning microscope with TCS SPE system (Leica DM2500. Leica Microsystems, Wetzlar, Germany) or Zeiss LSM900 with Airy-Scan technology. For quantification, 3 comparable regions of septum, ectoturbinate, and endoturbinate were studied using 3–5 mice per group. Cell counting and intensity measurement were performed in LAS X (Leica Microsystems), ZEN 3.0 (Carl Zeiss Microscopy GmbH, Oberkochen, Germany), and ImageJ. For image processing, Adobe Photoshop CS6 (Adobe Systems, CA, USA) was employed.

### 2.3. Heparanase Treatment

Before staining, slides were washed 3 × 10 min in PBS (pH 7.4), followed by incubation in heparanase-1 (5 µg/mL; Merck, Darmstadt, Germany, SAE0116) or PBS (as control, pH 5.6) for 24 h at 37 °C in a humid chamber. The next day, slides were washed 6 × 10 min in ice-cold PBS to stop the reaction. Afterwards, slides were processed for immunofluorescent staining as previously described.

### 2.4. Proximity Ligation Assay (PLA)

The use of PLA secondary antibodies derived against one host but conjugated to two different DNA oligonucleotides (PLUS and MINUS) enables detection of individual proteins. Due to the close proximity of PLA secondary antibodies when binding to single primary antibodies, conjugated and matching DNA oligonucleotides can be ligated to a circular DNA. Subsequent rolling circle amplification of the DNA and tagging with a fluorescence-conjugated DNA probe enables signal amplification and detection of individual proteins. In detail, slides were treated according to the immunofluorescence protocol using rabbit-derived anti-CXCL12 or anti-KRT5 antibodies and anti-rabbit PLA PLUS and MINUS antibodies (Duolink In Situ PLA Probe Anti-Rabbit PLUS; Duolink In Situ PLA Probe Anti-Rabbit MINUS, Sigma-Aldrich, St. Louis, MO, USA) as secondary antibodies. Slides were washed in TBS and incubated with ligation mixture (Duolink In Situ Detection Reagents Orange, Sigma-Aldrich, St. Louis, MO, USA) for 1 h at 37 °C. Ligated PLUS and MINUS strands were amplified in an isothermal rolling circle polymerase chain reaction (100 min, 37 °C) and tagged by Cyanine-3-conjugated probes (Duolink In Situ Detection Reagents Orange, Sigma-Aldrich, St. Louis, MO, USA). Additionally, PLA PLUS and MINUS secondary antibody controls were performed without applying primary antibody. Images were taken with a confocal laser scanning microscope Zeiss LSM900 (Carl Zeiss Microscopy GmbH, Oberkochen, Germany) with Airy-Scan technology.

### 2.5. PLA Signal Density Heatmap Generation

PLA images were processed using ZEN 3.0 (blue edition) (Carl Zeiss Microscopy GmbH, Oberkochen, Germany) software. The images of each z-stack were combined into an orthogonal projection. For all images, the background was reduced equally by adjusting the intensity values. PLA signal density heatmaps were generated using Fiji (ImageJ 1.53c) [41], applying the Lookup Table “Red Hot” and a Gaussian Blur filter with a Sigma radius of 20. For better orientation, the heatmap image was overlayed onto a high-background image showing the nuclei generated by using the “remove outliers” function of Fiji.

### 2.6. In Situ Hybridization

The fluorescent sense and antisense riboprobes for mouse *Ackr3* (GI 109562 [31] and mouse *Cxcr4* (GI 109563 [42]) were labeled with digoxigenin-uridine triphosphate (DIG-UTP, Perkin Elmer, MA, USA; for *Ackr3*) and fluorescein-12-UTP (Perkin Elmer, MA, USA, for *Cxcr4*). For radioactive in situ hybridization, sense and antisense riboprobes for mouse *Cxcl12α* (GI 12025675 [43]) were constructed using ^35^S-UTP (Perkin Elmer, MA, USA). Probe transcription and prehybridization were performed as previously described [16]. Sections were hybridized with 1 µg/mL fluorescent or radioactive riboprobe in hybridization buffer for 20 h at 60 °C in a humid formamide chamber, followed by washing in 2 × Tri-sodium-citrate-dihydrate (SSC) and 1 × SSC. Afterwards, sections were incubated in RNAse solution for 30 min at 42 °C and washed further in 1× and 0.2 × SSC.

For double fluorescent in situ hybridization, sections were incubated in blocking buffer for 1 h, followed by Anti-DIG-alkaline phosphatase (AP) fab fragments (1:500; Roche Deutschland Holding GmbH, Grenzach-Wyhlen, Germany; 11093274910) for 30 min. After washing with maleic acid, sections were stained with TSA TM Plus Cyanine 3 (1:50; NEL744001KT; Perkin Elmer, MA, USA) for 15 min in the dark to detect *Ackr3* mRNA first. In the second part, sections were washed twice in maleic acid, incubated in 3% H_2_O_2_ (in maleic acid) for 1 h, and rinsed again in maleic acid. Afterwards, sections were incubated in the second fab fragment cocktail (Anti-Fluorescein-AP fab fragments; 1:500; Roche Deutschland Holding GmbH, Grenzach-Wyhlen, Germany; 11426338910) 30 min at RT, washed with maleic acid, and stained with TSA TM Plus Fluorescein (1:50; NEL741E001KT; Perkin Elmer, MA, USA) in order to detect *Cxcr4* mRNA in the same slide. After washing with maleic acid, sections were mounted with Fluoromount-GTM (Thermo Fisher Scientific Germany Ltd. & Co. KG, Bonn, Germany). Images were performed using a confocal laser scanning microscope with TCS SPE system (Leica DM2500, Leica Microsystems, Wetzlar, Germany). For radioactive in situ hybridization, sections were dehydrated in isopropanol after washing in SSC. Dried slides were dipped into NTB-Emulsion (42 °C; IBS-Integra Biosciences AG, Zizers, CH; 10542844), dried for 24 h, and stored 4 weeks in the dark at 4 °C. Afterwards, coated sections were developed in warm Kodak^®^ Professional D-19 Developer (45 °C; Kodak, NY, USA; 1464593) and fixed in Carestream^®^ Kodak^®^ Processing Chemicals Kodak Fixer (Sigma Aldrich, MO, USA; P8307), counterstained with cresyl violet, and mounted with Entellan (Sigma Aldrich, MO, USA) [16]. Dark and bright field images were taken on a light microscope (Zeiss Axio Imager A1, Carl Zeiss, Oberkochen, Germany).

### 2.7. Immunosorbent Assay (ELISA)

Mice were decapitated and skin and lower jaw were removed. Pharyngeal nasal lavage protocol was adaped from [44]. A 24G catheter of a butterfly cannula (Braun, Melsungen, Germany, 42540748) was inserted carefully into the nasopharynx and choanae. A single dose of 700 µL ice-cold PBS was injected and mucus fluid was collected from the nostrils. Samples were concentrated in Micro Float-A-Lyzer^®^ Dialysis Device in Spectral/Gel™ Absorbent for 60 min. CXCL12 protein levels in mucus and OE samples were quantified using RayBio^®^ Mouse SDF-1α (CXCL12) ELISA Kit (RayBiotech, Peachtree Corners, GA, USA, ELM-SDFα). Lactoferrin protein levels were measured using Mouse LTF/LF (Lactoferrin) ELISA Kit (Elabscience^®^, Houston, TX, USA, E-EL-M0746). The ELISA assays were performed as recommended in the manufacturers’ instructions. Color change was detected on an iMark Microplate Reader (Biorad, CA, USA) at 450 nm wave length. ELISA assays were performed with n = 5 and 3 independent runs.

### 2.8. RNA Sequencing and Transcriptomic Analysis

Main olfactory epithelium was collected from P8 WT and *Mpz*-Cre;*Ackr3*^LoxP/LoxP^ mice. Two epithelia were pooled for each of three samples from the same genotype. RNA isolation was performed using Purelink RNA Mini Kit; RNA concentration and purity was determined by NanoDrop Lite (Thermo Fisher Scientific Germany Ltd. & Co. KG, Bonn, Germany). RNA integrity was controlled using denaturing agarose gel runs. Transcriptional sequencing was performed on NovaSeq6000 systems by CeGaT GmbH (Tübingen, Germany).

Un-normalized RNA read counts obtained from high-throughput RNA sequencing were tested for differential expression using the R Bioconductor software package DESeq2 [45]. The data set was processed following the provided standard workflow and filtered for an adjusted *p*-value cutoff of 0.05. To determine the histological localization and average expression, differentially expressed genes were identified in a published RNA single-cell-sequencing data set GSE169011 [46]. Up- (FC > 1.2) and downregulated (FC < 0.8) genes were further filtered for highest combinatory average expression in CXCR4-positive GBCs and INPs in contrast to all other cell types. Visualization was performed using the DotPlot function of the R software package Seurat V4 [47]. Gene ontology enrichment and network analysis was accomplished using ShinyGO 0.76.3 [48] for GO biological processes and an FDR cutoff of 0.05.

### 2.9. Statistical Analysis

Statistical analysis was performed in GraphPad Prism 5.01; data were represented as mean ± SEM. Data were tested for normal distribution and homogeneity. Statistical significance was set at * probability (*p*) < 0.05 and analyzed using One-Way ANOVA or Two-Way ANOVA, and Bonferroni post-hoc test or Student’s *t*-test.

## 3. Results

### 3.1. ACKR3 Is Expressed in Glia Cells of the OE

CXCL12/CXCR4-mediated signaling is regulated by the atypical chemokine receptor 3 (ACKR3), which modulates the extracellular CXCL12 concentrations by internalization and lysosomal degradation of CXCL12 [49]. In addition, CXCR4 signaling can be modulated through the formation of ACKR3-CXCR4 heterodimers. Since CXCR4 regulates proliferation and differentiation of GBCs [16], we investigated the role of ACKR3 for CXCR4 signaling in the OE. We started by analyzing young mice (postnatal day 8, P8), due to the high rate of neurogenesis at this age. In situ hybridization revealed *Ackr3* expression in the sustentacular cell layer of P8 animals (Figure 1A). The expression pattern did not change in adult animals. Co-hybridization of *Ackr3* with a *Cxcr4*-specific riboprobe showed that transcripts of both receptors do not co-localize. In compliance with the in situ hybridization result, expression of GFP under the control of the *Ackr3* promoter (*Ackr3*-GFP mice) showed labeling of sustentacular cells. The GFP signal was detected in the apical cell bodies of sustentacular cells and in the basal processes of sustentacular cells (Figure 1B), which span the complete epithelium (Figure 1D). Immunofluorescence labeling of ACKR3 and CXCR4 confirmed presence of both receptors in different cell types. ACKR3 was localized in the apical cytoplasm of sustentacular cells, whereas CXCR4 was localized in GBCs and immature neurons (Figure 1C). The specificity of our ACKR3 staining was validated by a virtually identical pattern being obtained by anti-hemagglutinin (HA) staining in mice with knock-in alleles for HA epitope-tagged ACKR3 [25] (Figure 1E). High-power confocal microscopy showed that ACKR3-positive processes of presumptive sustentacular cells surrounded CXCR4-labeled dendrites and cell bodies (Figure 1F), ruling out that ACKR3 modulates CXCR4 by heterodimer formation. Together, these findings show the presence of ACKR3 in CXCR4-negative sustentacular cells in the OE.

### 3.2. ACKR3 in Sustentacular Cells Regulates CXCR4 Activation

CXCL12 mostly localized to HBCs in WT mice with weak staining throughout the epithelial layer (Figure 2A). In addition, horizontal sections through the cell bodies of sustentacular cells revealed a weak CXCL12 signal in the cytoplasm (Figure 2B). To confirm the presence of CXCL12 in the OE, and to test whether the chemokine is also present in the nasal mucus, we next performed enzyme-linked immunosorbent assay (ELISA) for CXCL12. Lactoferrin, a secreted iron-binding, antimicrobial protein in body secretions, which is an important component of the non-specific immune system, served as positive control (Figure 2C). We detected a high CXCL12 concentration in homogenates of the OE and a low concentration in nasal lavage fluid. Lactoferrin was detected at a concentration of ~1 µg/µL in the nasal mucus, which corresponds to concentrations reported for human tears (2 µg/µL) and saliva (0.008 µg/µL) [50]. Taken together, these findings show that CXCL12 is present in the OE and nasal mucus.

To examine if ACKR3 regulates the CXCL12 concentration in the OE, we developed a myelin protein zero (MPZ) promoter-driven knockout mouse model of sustentacular cell ACKR3 (*Mpz*-Cre;*Ackr3*^LoxP/LoxP^). *Ackr3* mRNA and ACKR3 protein were abolished in the OE of these mutants (Appendix A). This confirms *Ackr3* ablation and shows that *Ackr3* is normally expressed in sustentacular cells. Due to the fact that ACKR3 is canonically involved in scavenging CXCL12, we examined CXCL12 levels in the OE of *Mpz*-Cre;*Ackr3*^LoxP/LoxP^ mice (P8). Localization of CXCL12 in HBCs was not altered in *Mpz*-Cre;*Ackr3*^LoxP/LoxP^ mice, but we noticed a slightly increased CXCL12 signal in the apical OE layers (Figure 2D), which would be consistent with the hypothesis that ACKR3 in sustentacular cells acts as scavenging receptor and regulates the CXCL12 concentration in the OE. A similar finding was obtained in ACKR3-ST/A mice (Figure 2E), a model of impaired ACKR3-mediated CXCL12 uptake due to alanine mutations in the highly conserved C-terminal ^350^SETE^353^ phosphorylation motif of ACKR3 [25].

Immunofluorescence stainings showed CXCL12 in the sustentacular and neuronal layer of the OE, but the staining was relatively faint. To confirm the increase in the epithelial CXCL12 signal in the OEs of *Mpz*-Cre;*Ackr3*^LoxP/LoxP^ and ACKR3-ST/A mice, we performed proximity ligation assay (PLA) [51], a technique that was used recently to analyze the distribution of CXCL12 in the bone marrow [52]. For visualization, we generated concentration heatmaps from PLA dot locations (Figure 2G and Appendix A). The comparison between WT and *Mpz*-Cre;*Ackr3*^LoxP/LoxP^ mice revealed a distinct increase in the CXCL12 concentration in the apical part of the OE (Figure 2F,G). This is likely due to absent CXCL12 scavenging, since the scavenging-impaired ACKR3-ST/A mutant exhibited a similar effect on CXCL12 localization (Figure 2F,G). As a control, PLA labeling of CXCL12 in CXCL12-overexpressing mice also showed increased PLA signals (Appendix A). We did not find PLA signals in the HBC layer, most likely due to the well-known inhibition of DNA polymerase (necessary for rolling circle amplification of PLA signals) by heparan sulfate (HS) proteoglycans [53], which can be detected on HBCs ([54] and Appendix A). Attempts to remove HS from tissue sections by heparanase digestion unfortunately resulted in a complete loss of the PLA signal. Taken together, increased amounts of CXCL12 in the apical OE of ACKR3-deficient mice, together with ACKR3 localization in intracellular vesicular structures in the sustentacular cells, shows that ACKR3 plays a role in CXCL12 scavenging in the OE.

### 3.3. CXCL12 Scavenging by ACKR3 Regulates CXCR4 Signaling

Next, we investigated how CXCL12 scavenging by ACKR3 affects activation of CXCR4 and downstream signaling. CXCR4 activation can be detected with the anti-CXCR4 antibody UMB-2 which recognizes the C-terminal CXCR4 epitope only when the ^346^SSS^348^ cluster is not phosphorylated [55]. Since phosphorylation occurs upon receptor stimulation, the UMB-2 signal is lost in activated CXCR4. As expected, the overall UMB-2 signal was markedly reduced in *Mpz*-Cre;*Ackr3*^LoxP/LoxP^ and ACKR3-ST/A mice, indicating massive phosphorylation and thereby activation of CXCR4 (Figure 3A). Higher magnification pictures moreover showed different distribution of non-phosphorylated CXCR4 (Figure 3B). In WT mice, non-phosphorylated CXCR4 was present on the plasma membrane and in intracellular clusters. In ACKR3-ST/A and *Mpz*-Cre;*Ackr3*^LoxP/LoxP^, non-phosphorylated receptors were no longer detectable on the cell surface, while the amount of intracellular clusters increased. Formation of CXCR4 clusters is known to be associated with exposure to high levels of CXCL12 [56,57]. Notably, CXCR4-positive spots were positioned close to the γ-tubulin-labeled microtubule-organizing center (MTOC) and were only found in post-mitotic neuronal progenitor cells, not in HBCs or dividing GBCs (Appendix A). In addition to the differences in protein localization, total amounts determined by staining intensities were altered in the same way (Figure 3C).

Considering that excessive CXCL12 levels cause CXCR4 activation and degradation [16,30], we next examined CXCR4 expression in the OE. Both mutants, *Mpz*-Cre;*Ackr3*^LoxP/LoxP^ and ACKR3-STA mice, exhibited a reduction in the number of CXCR4-positive cells (Figure 3D,E), as expected. Moreover, the distribution of CXCR4 was altered when ACKR3-mediated scavenging was inhibited. WT mice showed CXCR4 labeling on the plasma membrane of GBCs and some immediate neuronal progenitor cells extend CXCR4-positive dendrites towards the surface of the epithelium, whereas dendritic localization of CXCR4 was nearly abolished in ACKR3-deficient (ACKR3-ST/A and *Mpz*-Cre;*Ackr3*^LoxP/LoxP^) animals (Figure 3D). Similar effects on the expression of CXCR4 were present in adult animals. CXCL12 binding to CXCR4 inhibits adenylyl cyclase via Gαi [58] and activates phosphoinositide-3 kinase (PI3K) via the Gβ/Gγ dimer [59]. PI3K activates p70S6K (p70 ribosomal S6 kinase), a member of the S6K family of serine/threonine kinases, which phosphorylates several residues of the S6 ribosomal protein. Consistent with CXCR4 downregulation, we observed reduced phosphorylation of S6 in the OE of *Ackr3*-deficient mice (Figure 3F,G). Again, we saw a similar phenotype in ACKR3-ST/A mice, showing that the absence of scavenging leads to downregulation of CXCR4 signaling (Figure 3F,G). Altogether, our data establish that scavenging by ACKR3 regulates CXCL12 availability in the OE.

### 3.4. CXCL12 Scavenging by ACKR3 Regulates Stem Cell Proliferation and Neurogenesis

In order to further analyze the impact of deficient CXCL12 scavenging in the OE, we performed transcriptome analysis of mRNA from OE of P8 WT and *Mpz*-Cre;*Ackr3*^LoxP/LoxP^ mice. To identify the genes specifically regulated in CXCR4-positive GBCs and INPs, we mapped the regulated genes to published single-cell sequencing data [46]. The mapping matched 56 of the downregulated genes to GBCs and INPs (Figure 4A), while only two of the upregulated genes were found in these cell types (*Larp4, Uchl5*). GO analysis of the differentially expressed genes revealed that genes involved in cell proliferation or cell cycle progression were enriched (Figure 4B,C). Moreover, network analysis demonstrated that most identified GO terms are strongly connected (Figure 4C). Among the downregulated genes is MKI67, an essential component of the perichromosomal layer and a marker for the condensed chromosomes in mitotic cells. Staining of proliferating cells in the OE with MKI67 antibodies confirmed this observation (Figure 4D,E), implying that the proliferation of CXCR4+ GBCs is inhibited by high CXCL12 levels in the microenvironment. Altered PI3K activity correlates well with altered cell proliferation. A reduction in cell proliferation was confirmed by analyzing the expression of MCM2 (Minichromosome Maintenance 2), a proliferation marker involved in the initiation of DNA replication (Figure 4F,G).

### 3.5. CXCL12 Availability Regulates CXCR4-Dependent Neurogenesis

Another important aspect revealed by transcriptome analysis is that half (27 of 55) of the downregulated genes in GBCs of *Mpz*-Cre;*Ackr3*^LoxP/LoxP^ mice are known to be involved in neurogenesis (marked red in Figure 4A). For example, downregulated *Mad1l1* (mitotic arrest deficient 1 like 1) contributes to morphogenesis during brain development by regulating the integrity of the Golgi complex [60]. Interestingly, we detected an altered morphology of the Golgi apparatus in olfactory neurons of mice with deficient CXCL12 scavenging. We moreover found regulation of the REST corepressor 2 (*Rcor2*, also CoREST), which plays a key role in epigenetic regulation of cortical development [61]. Interestingly, of the 55 downregulated transcripts detected in the *Mpz*-Cre;*Ackr3*^LoxP/LoxP^ OE, 13 (*Hes6, Insm1, Rcor2, Ncapd2, Knch8, Gpc2, Cnpy, Sbk1, Hr, Fezf1, Thsd7b, Gng8, Robo3*) were concordantly downregulated in neuronal progenitor cells of (embryonic) *Nes*-Cre;*Rcor1*^LoxP/LoxP^;*Rcor2*^LoxP/LoxP^ mice [62]. Depletion of *Rcor2* in neurogenic stem cells results in reduced proliferation and downregulation of MKI67 [61], also downregulated in the OE transcriptome of *Mpz*-Cre;*Ackr3*^LoxP/LoxP^ mice (Figure 4). RCOR2 is well known to recruit the lysine-specific histone demethylase 1A (*Kdm1a*, also LSD1) to regulatory genomic regions, and LSD1 expression in GBCs and INPs plays a role for maturation of OSNs [63,64]. Immunofluorescence staining showed that *Mpz*-Cre;*Ackr3*^LoxP/LoxP^ mice and ACKR3-ST/A mice showed increased staining of LSD1 (Figure 5A,E). We therefore analyzed markers of neuronal differentiation. Growth-associated protein 43 (GAP43), a marker for immature olfactory neurons and known to regulate neurite outgrowth [65], was localized on the plasma membrane and clustered in the cell bodies of numerous immature neurons present in P8 animals (Figure 5B). *Mpz*-Cre;*Ackr3*^LoxP/LoxP^ and ACKR3-ST/A mice showed reduced labeling (Figure 5B). The high density impedes exact counting of immature neurons, but staining intensities in confocal pictures were significantly different for ACKR3-ST/A mice (Figure 5F). The structurally related Myristoylated Alanine-Rich C Kinase Substrate (MARCKS), also described as a marker of neuronal growth [66], showed similar alterations in staining intensities depending on CXCL12 availability (Figure 5C,G). The fact that the observed differences were very similar between *Mpz*-Cre;*Ackr3*^LoxP/LoxP^ and ACKR3-ST/A mice most likely reflects altered CXCR4 signaling, caused by changes in the CXCL12 concentrations. Since both GAP43 and MARCKs are involved in dendrite growth, altered dendritic localization of both shows that CXCR4-CXCL12 signaling alters neuronal development. Due to an overall declining rate of neurogenesis during postnatal development, reduced numbers of GAP43-positive immature neurons were observed in 8W old WT animals (Appendix A). However, also adult *Mpz*-Cre;*Ackr3*^LoxP/LoxP^ animals showed a reduction in the number of GAP43-positive immature neurons compared to WT, similar to P8 animals.

To complement the above observations, we analyzed the number of mature neurons by staining of olfactory marker protein (OMP), a well-established marker of mature olfactory neurons [67]. The number of OMP-positive cells was higher in mice with impaired CXCL12 scavenging (ACKR3-ST/A and *Mpz*-Cre;*Ackr3*^LoxP/LoxP^) (Figure 5D,H). In adult mice (8W), we measured a slightly increased thickness of the layer of mature (OMP-positive) neurons in *Mpz*-Cre;*Ackr3*^LoxP/LoxP^ mice, but the difference was not significant. To this end, increased concentrations of CXCL12 reduced the immature cell population and increased the OMP-positive mature cell population during postnatal development. Tight regulation of CXCL12 is therefore important for sustained neurogenesis in the olfactory stem cell niche.

### 3.6. Cxcl12 Expression in Sustentacular Cells Is Necessary for Proper Regulation of Neurogenesis

After demonstrating ACKR3-mediated tight regulation of CXCL12 levels in the OE, we sought to identify where *Cxcl12* is expressed. The lamina propria of young mice (P8) expressed large amounts of *Cxcl12* mRNA, whereas *Cxcl12* expression was somewhat lower in older mice (8W) (Figure 6A). In addition to the lamina propria, we detected *Cxcl12* mRNA apically in the sustentacular cell layer in P8 animals. Given that CXCL12 levels in the apical OE were found to be relevant for neurogenesis, we assessed if CXCL12 emanating from sustentacular cells exerts a functional impact on CXCR4 signaling. For targeted *Cxcl12* deletion, we generated *Mpz*-Cre;*Cxcl12*^LoxP/LoxP^ mice. The localization of CXCL12 in HBCs was not altered in these mice, indicating that CXCL12 in HBCs is not derived from sustentacular cells (Appendix A). In addition, the CXCL12 immunofluorescence detection in the apical portion of the OE was very similar (Appendix A), which may be due to the fact that a reduction in the scarce CXCL12 labeling in WT mice cannot be detected by immunofluorescence labeling. Thus, we determined whether there was a reduction in the activation of CXCR4. While CXCR4 was localized in cell bodies of basal cells and along the dendrites of immature neurons in WT mice, *Mpz*-Cre;*Cxcl12*^LoxP/LoxP^ mice showed an increase in CXCR4 membrane localization, which was particularly obvious in the apical part of the OE (Figure 6B). The number of CXCR4-positive cells was increased slightly, consistent with reduced ligand-induced degradation, but the difference did not reach statistical significance (Figure 6C). Markedly increased labeling with the UMB-2 antibody in mutant mice then confirmed that CXCL12 deletion in the sustentacular cells of *Mpz*-Cre;*Cxcl12*^LoxP/LoxP^ mice resulted in reduced CXCL12–CXCR4 signaling (Figure 6D–F). UMB-2 positive punctae were hardly ever observed, indicating that their presence depended on CXCL12. Lack of CXCL12 secretion from sustentacular cells led to particularly strong labeling of apical dendrites (Figure 6E). In addition, confocal microscopy of horizontal sections through the layer of the dendritic knobs revealed little labeling of CXCR4 or non-phosphorylated CXCR4 in WT mice, but abundant labeling in *Mpz*-Cre;*Cxcl12*^LoxP/LoxP^ mice (Figure 6G). Since MPZ is weakly expressed in ensheathing cells of the lamina propria, we performed *Cxcl12* in situ hybridization in *Mpz*-Cre;*Cxcl12*^LoxP/LoxP^ mice and found that *Cxcl12* was still expressed in the lamina propria (Appendix A). Thus, increased CXCR4 labeling in the OE of *Mpz*-Cre;*Cxcl12*^LoxP/LoxP^ mice is not due to extraepithelial *Cxcl12* deletion. Together, these results show that sustentacular-cell-derived CXCL12 contributes to CXCR4 activation in the stem cell niche and the apical dendrites of INPs and iOSNs.

### 3.7. CXCL12 Detected in HBCs Is Derived from the Lamina Propria

In addition to sustentacular cells, a large amount of *Cxcl12* mRNA was synthesized in the lamina propria of P8 and adult (8W) animals (Figure 6A and Figure 7A). Moreover, histological assessment by in situ hybridization revealed a massive expansion of *Cxcl12* mRNA in the lamina propria 3 days after injury of the OE by intraperitoneal methimazole injection [68], coinciding with marked activation and proliferation of HBCs (Figure 7B). However, CXCL12 protein was detected in activated HBCs building the OE (Figure 7C,D). At steady state, CXCL12 protein co-localized with the HBC marker intercellular adhesion molecule 1 (ICAM1), but not with the sustentacular cell marker Keratin 8 (KRT8) in the basal part of the OE (Figure 7D).

We therefore aimed to test if CXCL12 originating from the lamina propria gains access to the OE and might thus explain the CXCL12 signal on *Cxcl12*-negative HBCs. CXCL12 is displayed on cell surfaces in association with heparan sulfate (HS), part of the extracellular domain of syndecans. Syndecans are highly conserved type I transmembrane proteoglycans that are present on the basolateral surface of the OE [54]. Moreover, Syndecan 1 (SDC1) was abundant at the early phase of methamizol-induced neurogenesis (Figure 7E), coinciding with CXCL12 accumulation in activated HBCs. Moreover, CXCL12 in HBCs co-localized with HS and with SDC1, indicating that CXCL12 is indeed present in the extracellular matrix of HBCs (Figure 7E).

To verify that CXCL12 on HBCs was not derived from low-level expression in HBCs themselves, we knocked out *Cxcl12* in HBCs using *Keratin 14* (*Krt14)*-Cre, which mediates genetic recombination in HBCs and sustentacular cells, but not in GBCs or neurons (Appendix A). Immunolabeling showed that CXCL12 was still present in HBCs of *Krt14*-Cre;*Cxcl12*^LoxP/LoxP^ mice and did not show obvious differences in labeling intensities compared to controls (Figure 7F,G). Moreover, CXCL12 on HBCs was not altered in *Mpz*-Cre;*Cxcl12*^LoxP/LoxP^ mice (Figure 6), proving that HBC CXCL12 is not derived from sustentacular cells. As a result of these findings, it appears that the CXCL12 signal on HBCs is neither caused by low-level *Cxcl12* expression in HBCs nor derived from the basal processes of sustentacular cells. With *Cxcl12* expression absent in the other intraepithelial cell types, this leaves the lamina propria as the most likely CXCL12 source for HBCs and suggests that *Cxcl12*-negative HBCs accumulated CXCL12.

### 3.8. CXCL12 Accumulates in the Extracellular Matrix Surrounding HBCs

Next, we aimed to test by a genetic approach whether CXCL12 accumulation in HBCs indeed depends on HS. Since knock-out of enzymes that generate HS is embryonically lethal, we used mice with deficiency in α-l-iduronidase (IDUA), a lysosomal enzyme that hydrolyses alpha-L-iduronosidic linkages. Reduced lysosomal degradation then leads to accumulation of HS and dermatan sulfates and presents as Hurler disease (Mucopolysaccharidosis I, MPSI-H) [69]. The bone marrow of *Idua*-deficient mice contains significant amounts of HS in the extracellular matrix, elevated CXCL12 levels, and homing defects of CXCR4-positive hematopoietic stem cells [70]. We therefore anticipated CXCL12–CXCR4 signaling to be increased in the OE of *Idua*-deficient mice. Considering that heparan sulfate accumulates over the lifetime, we analyzed adult (8W) animals instead of the P8 animals on which most other experiments were carried out due to high neurogenesis rates. We first stained for CXCL12 and found only a slight, non-significant increase in the CXCL12 signal in *Idua^−/−^* mice (Figure 8A,B). Absence of a detectable difference in CXCL12 immunofluorescence staining does not rule out an effect of HS in CXCL12 accumulation. We therefore further examined if the increase in HS in *Idua*^−/−^ mice was associated with CXCL12 accumulation by analyzing CXCR4 activation with the UMB-2 antibody. As expected, we detected markedly weaker labeling with the UMB-2 antibody in *Idua*^−/−^ mice, indicating over-activation of CXCR4 (Figure 8C,F). Since activated CXCR4 is internalized and subjected to lysosomal degradation, sites of active CXCR4 signaling often exhibit a decrease in CXCR4. Consistently, the OE of *Idua*^−/−^ mice also presented with clearly reduced CXCR4 expression (Figure 8C,F). Increased CXCR4 signaling was substantiated by showing a reduced number of GAP43-positive immature neurons in *Idua*^−/−^ mice (Figure 8D,G), which was similar to 8W old *Mpz*-Cre;*Ackr3*^LoxP/LoxP^ mice, but distinct from *Mzp*-Cre;*Cxcl12*^LoxP/LoxP^ mice. These results show that the number of immature neurons was consistently decreased when CXCL12 levels were higher. The differences in the number of mature neurons were smaller, but mice with increased CXCL12 (*Idua*^−/−^*, Mpz*-Cre;*Ackr3*^LoxP/LoxP^) were significantly different from mice with reduced CXCL12 (*Mpz*-Cre;*Cxcl12*^LoxP/LoxP^) (Figure 8H). Collectively, our findings indicate that extracellular-matrix components surrounding HBCs play a role in CXCL12 presentation in the olfactory stem cell niche (Figure 8E).

## 4. Discussion

Understanding stem cell niches is crucial to gain deeper insight into tissue homeostasis and regeneration. We show here how new neurons in the olfactory epithelium arise from the collective interactions of glial cells and the extracellular matrix. Tight regulation of the CXCL12 concentration by sustentacular cells is required to form the appropriate environment for the development of olfactory neurons. In the sustentacular cells, ACKR3 expression fine-tunes CXCL12 availability, constituting the first description of ACKR3-mediated scavenging in the post-developmental phase. We complement these findings by showing that HS-dependent accumulation of CXCL12 in HBCs is required for presentation of CXCL12 in the olfactory stem cell niche.

Cells of the lamina propria are responsible for a significant portion of CXCL12 in the olfactory mucosa, based on in situ hybridization. Lamina propria-derived CXCL12 could be synthesized by olfactory glia, the olfactory ensheathing cells [71], but also from other cells in the lamina propria such as fibroblasts, mesenchymal stem cells, or cells from the Bowman glands. MPZ is a myelin sheet protein of peripheral nerves which is expressed by olfactory ensheathing cells, although olfactory axons are not myelinated. Deletion of *Cxcl12* with *Mpz*-Cre did not completely abolish *Cxcl12* mRNA expression in the lamina propria, indicating that other cells also express *Cxcl12*. In situ hybridization cannot rule out some effect on the expression level, but CXCL12 accumulation on HBCs was not altered, suggesting that the amount of lamina propria-derived CXCL12 is not critical for OE CXCL12 supply. Although CXCL12 expression in astrocytes is increased during demyelination [72,73], olfactory neurons are not myelinated, and we were not able to detect substantial differences in the structure of axon bundles and glomeruli.

Matrix components provide localizing niche elements that can influence stem cell pools, and HS binding is important for the proper in vivo function of CXCL12 [29]. Here, we provide evidence that glycosaminoglycans presented by HBCs are crucial elements in chemokine presentation in the OE. Within the *Idua^−/−^* mouse with excess HS in extracellular locations [70], over-stimulation of CXCR4 in the stem cell niche leads to increased differentiation of cells of the neuronal linage. The situation is somewhat reminiscent of the bone marrow, where HS-bound CXCL12 regulates CXCR4-dependent signaling in hematopoietic stem and progenitor cells [28,70]. HBCs are reserve stem cells, which are quiescent and contribute little to the normal cell turnover during homeostatic neurogenesis [74]. During chronic inflammation, HBCs release cytokines and chemokines such as CCL19, CCL20, and CXCL10 to regulate inflammatory cell recruitment and local proliferation [12]. We identify here a previously unrecognized role of HBCs in orchestrating neurogenesis by accumulation of CXCL12 from the lamina propria.

CXCL12 is also expressed within the epithelium by sustentacular cells in addition to being taken up from the lamina propria. Expression of chemokines adds a new function for sustentacular cells, which are known to phagocytose dead and dying cells [75] and to eliminate noxious substances from the mucus [76,77]. More recently, sustentacular cells have been established as main site of SARS-CoV-2 infection [78]. Interestingly, infection-associated gene expression changes in sustentacular cells appear to reflect a response to inflammatory signaling, including upregulation of chemokines [79,80]. We show here that sustentacular cells release chemokines not only in response to (viral) infections, but also constitutively to regulate stem cell fate and tissue morphogenesis.

Different cell types continuously produce CXCL12 [81], a crucial chemokine in many homeostatic processes. Nevertheless, CXCL12 activity is regulated and tightly controlled. We demonstrate here the regulation of chemokine availability in the olfactory stem cell niche by ACKR3-mediated scavenging. Deletion of ACKR3 in sustentacular cells resulted in marked CXCR4 internalization, much like the situation in CXCL12-overexpressing mice [16]. In addition, CXCR4 internalization was very similar in mice with *Ackr3* knock-out in sustentacular cells and in phosphorylation-deficient HA-ACKR3-ST/A mice, verifying deficient scavenging as an underlying cause. In the developing nervous system, *Ackr3* deletion causes overstimulation and loss of CXCR4 in a cell-autonomous manner [30,31,32,33]. In the OE, ACKR3 expression is restricted to glia cells, thereby acting in a non-cell-autonomous manner on neuronal stem cells. Not least, we describe here the first post-developmental process involving ACKR3-mediated CXCL12 scavenging. Remarkably, the effects of *Ackr3* deletion, resulting in increased CXCL12 concentrations, and *Cxcl12* deletion in sustentacular cells were polar opposites. Both influence CXCR4 expression in stem cells; *Ackr3* deletion leads to reduced, and *Cxcl12* deletion leads to enhanced, CXCR4 abundance on the membranes of stem and progenitor cells.

By increasing the amount of extracellular CXCL12 in the apical part of the OE, absence of ACKR3 leads to altered CXCR4 signaling in stem cells and neuronal progenitor cells. CXCR4-dependent proliferation of stem cells seemed to involve RCOR2 (CoREST) downregulation. The CoREST complex acts as a repressor of gene expression and regulates neuronal gene expression, thereby interfering with neuronal differentiation via REST (RE1 silencing transcription factor) [82]. We also observed an increase in LSD1, an RCOR binding partner, in GBCs and INPs. Recruitment of LSD1 to specific genomic regions is known to play a role in the maturation of OSNs [63,64], and we saw an increased maturation of neuronal progenitors in *Ackr3*-deficient mice. Thereby, ACKR3 indirectly influences CXCR4 signaling by titrating the amount of the CXCR4 ligand CXCL12.

CXCL12 as extrinsic signal in the olfactory niche regulates the development of neuronal progeny and thereby the process of adult neurogenesis. The fact that more CXCL12 decreases the number of immature neurons, and concurrently increases the number of mature olfactory neurons, indicates that it serves to set the tempo of adult neurogenesis and facilitates cellular development. Neurogenesis in the intact and damaged OE requires a tight level of regulation to avoid imbalances in tissue size or composition, which are actually rare given the low numbers of neuroblastoma-like forms of cancer in the nose. The rate of olfactory neurogenesis might therefore be tied to the regulation CXCL12.

Larger or smaller than normal numbers of OSNs may also change odor perception, which would have to be investigated in a follow-up study. Through the use of mice of many different genotypes (summarized in Appendix A), this study shows tight regulation of CXCL12 during olfactory neurogenesis; however, this study is limited by the number of animals included per genotype.

## 5. Conclusions

Our results show that optimization of the ligand concentration is not only required for directional cell migration, but also plays a role in the differentiation of stem cells. In the olfactory niche, CXCL12 is an extrinsic signal that regulates the development of neuronal progeny and sets the pace of adult neurogenesis. There is a striking precision in the timing of olfactory neurogenesis, as evidenced by the sparseness of neuroblastoma-like cancers. CXCR4-expressing proliferative GBCs may be particularly sensitive to increased CXCL12 levels, requiring non-cell-autonomous ACKR3-mediated scavenging to ensure continuous CXCL12 responsiveness. We thereby provide the first clear example of a role for ACKR3-mediated CXCL12 scavenging in tissue homeostasis beyond developmental processes. Not least, a better understanding of the neurogenic niche within the OE may help to establish proper in vitro conditions to optimize growth and expansion of olfactory-tissue-derived stem cells in culture. CXCL12-producing olfactory ensheathing cells may hold promise in cell-based therapies for spinal cord injuries, as might CXCR4-expressing neuronal stem cells for treating neurodegenerative conditions. The lack of success with these approaches in clinical trials may be attributed to the lack of a favorable microenvironment after transplantation, with CXCL12 being an example of a neurotrophic molecule that needs a balanced concentration for optimal signaling.

## Figures and Tables

**Figure 1 cells-12-02164-f001:**
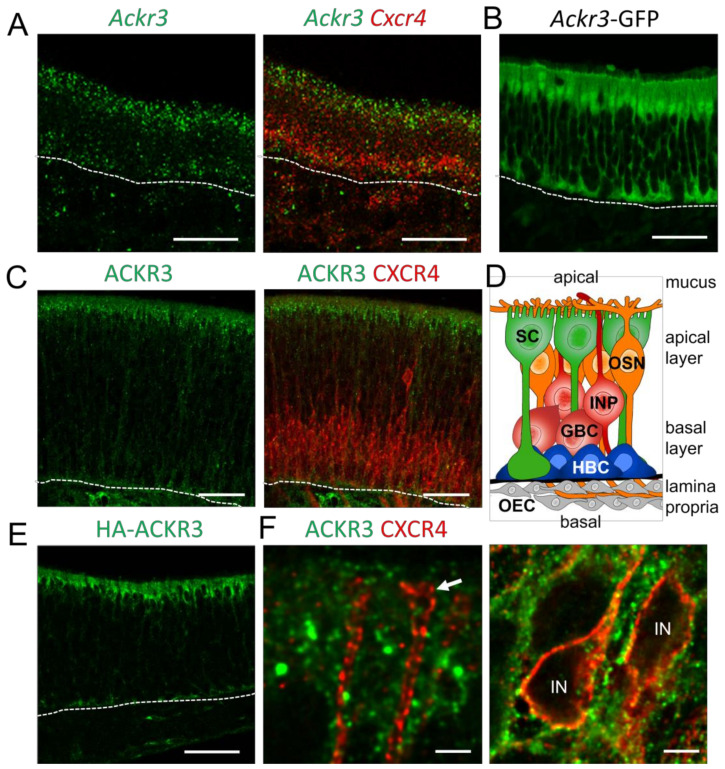
ACKR3 is not co-expressed with CXCR4 in the OE (**A**) Fluorescence in situ hybridization of *Ackr3* (green) with *Cxcr4* (red) showing expression of *Ackr3* in sustentacular cells of the OE (P8). *Cxcr4* and *Ackr3* mRNAs are not co-expressed. (**B**) Expression of GFP under the control of the *Ackr3* promoter (*Ackr3*-GFP transgenic mice) confirms *Ackr3* expression in sustentacular cells (P8). (**C**) Immunofluorescence for ACKR3 (green) and CXCR4 (red) showing absence of co-localization (P8). (**D**) Schematic overview of the cellular composition of the OE. Cell bodies of sustentacular cells (SC, green) build the apical cell layer and the cells extend to the basal lamina. Cell bodies of mature olfactory sensory neurons (OSN, orange) are located directly beneath the sustentacular cells; OSNs extend an apical dendrite with sensory cilia, which are in contact with the mucus covering the epithelium and an axon projecting through the lamina propria towards the olfactory bulb. The basal layer of the epithelium is composed of immature progenitors (IP, red) and globose basal cells (GBC, red), the proliferating stem cells generating the sensory neurons. Below the GBC layer, quiescent horizontal basal cells (HBCs, blue) are sitting on top of the basal lamina (thick black line). The lamina propria below the basal lamina contains olfactory ensheathing cells (OEC, grey) and other cells. (**E**) Knock-in mouse expressing HA-tagged ACKR3 (HA-ACKR3) labeled with anti-HA antibodies. ACKR3 is expressed in sustentacular cells (P8). (**F**) High magnification pictures of single confocal sections showing absence of co-localization of ACKR3 (green) and CXCR4 (red). Arrow points towards a dendritic ending; IN are cell bodies of immature neurons. Dotted lines represent the basal lamina. Scale bars 50 µm (**A**–**D**), 2 µm (**E**), 50 µm (**F**).

**Figure 2 cells-12-02164-f002:**
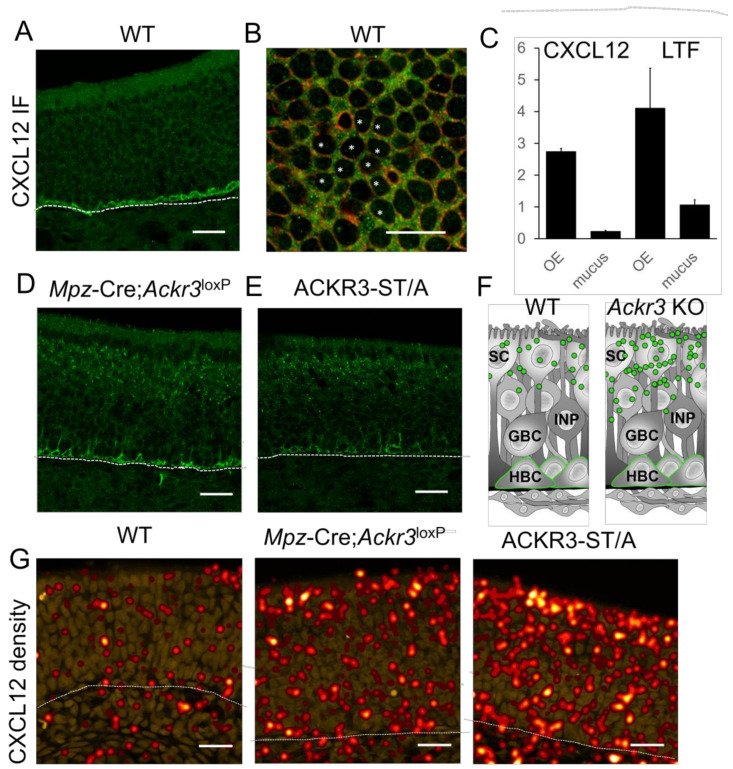
ACKR3 regulates the amount of CXCL12 in the OE. (**A**) Localization of CXCL12 in HBCs by standard IF in WT. (**B**) Horizontal section at the level of the sustentacular cell nuclei (not labeled, large black spaces, some are marked by white asterisk). Some CXCL12 (green) is present in the apical cytoplasm of the sustentacular cells expressing KRT8 (red). (**C**) Quantification of CXCL12 and lactoferrin (LTF) protein concentration in the OE and mucus samples of WT (6W) mice by ELISA. CXCL12 protein is abundant in the OE and also detectable in the mucus (n = 3 independent experiments, 4 animals per experiment, 3 technical replicates, error bars represent SEM). (**D**) In *Mpz*-Cre;*Ackr3*^LoxP/LoxP^ and (**E**) ACKR3-ST/A mice, some CXCL12 labeling can be observed in the apical OE. (**F**) Schematic overview of CXCL12 (green) localization in sustentacular cells (SC) and HBCs in the OE of WT and *Ackr3*-deficient mice. (**G**) OE sections of WT, *Mpz*-Cre;*Ackr3*^LoxP/LoxP^, and ACKR3-ST/A mice stained for CXCL12, densities were calculated from PLA signals. The amount of CXCL12 is much higher in mice not expressing ACKR3 or expressing scavenging-defective ACKR3-ST/A. Scale bars 20 µm (**A**,**D**,**E**,**G**); 10 µm (**B**).

**Figure 3 cells-12-02164-f003:**
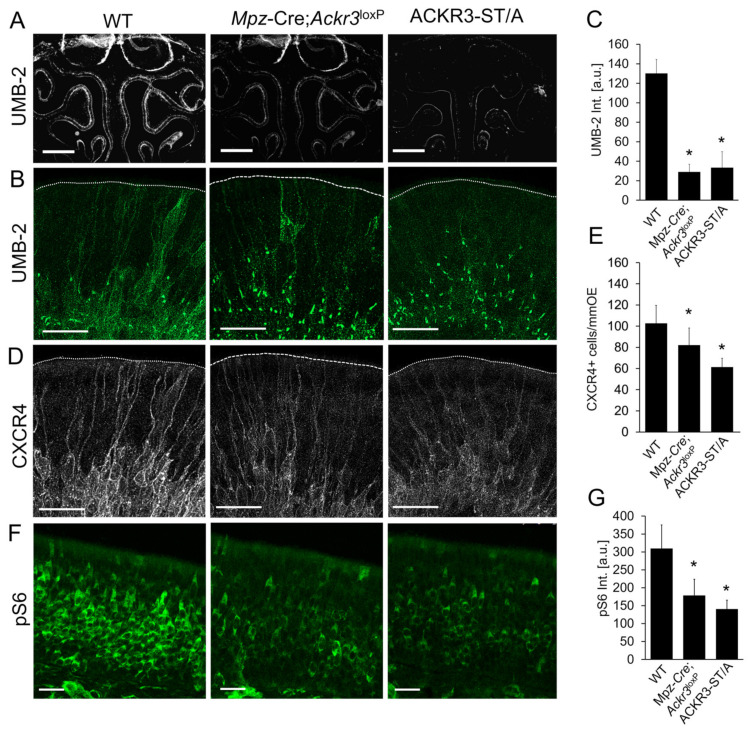
ACKR3 regulates CXCR4 signaling in olfactory stem cells. (**A**) Low-magnification overview pictures of UMB-2 (CXCR4 non-phosphorylated) immunolabeling in WT, *Mpz*-Cre;*Ackr3*^LoxP/LoxP^, and in ACKR3-ST/A mice (P8); shown are pictures taken with a stereomicroscope. (**B**) High-resolution Airyscan microscopy of UMB-2 (CXCR4 non-phosphorylated) immunolabeling in ACKR3-ST/A and in *Mpz*-Cre;*Ackr3*^LoxP/LoxP^ mice (P8), showing that activation of CXCR4 in dendrites and localization of non-phosphorylated CXCR4 depends on CXCL12 availability. CXCR4 in ACKR3-ST/A mice and *Mpz*-Cre;*Ackr3*^LoxP/LoxP^ was present mostly on intracellular clusters. Shown are projections of confocal sections (16 µm); dotted lines delineate the apical surface of the OE. (**C**) Quantification of UMB-2 staining intensities in cryosections (n = 3 animals per group, Student’s *t*-test, error bars represent SEM, * *p* < 0.05). (**D**) CXCR4 immunolabeling in the apical part of the OE shows reduced labeling of dendrites of progenitor cells in ACKR3-ST/A and *Mpz*-Cre;*Ackr3*^LoxP/LoxP^ mice. Shown are projections of confocal sections of P8 mice; Airyscan microscopy, total thickness 16 µm, dotted lines delineate the apical surface of the OE. (**E**) Quantification of CXCR4-positive cells in cryosections (n = 3 animals per group, Student’s *t*-test, error bars represent SEM, * *p* < 0.05). (**F**) Immunolabeling of phosphorylated S6 (pS6) in *Mpz*-Cre;*Ackr3*^LoxP/LoxP^ mice and ACKR3-ST/A mice (P8), showing reduced CXCR4 signaling. (**G**) Quantification of pS6 staining intensities in cryosections (n = 3 animals per group, Student’s *t*-test, error bars represent SEM, * *p* < 0.05). Scale bars 500 µm (**A**); 20 µm (all pictures in (**B**,**D**,**F**)).

**Figure 4 cells-12-02164-f004:**
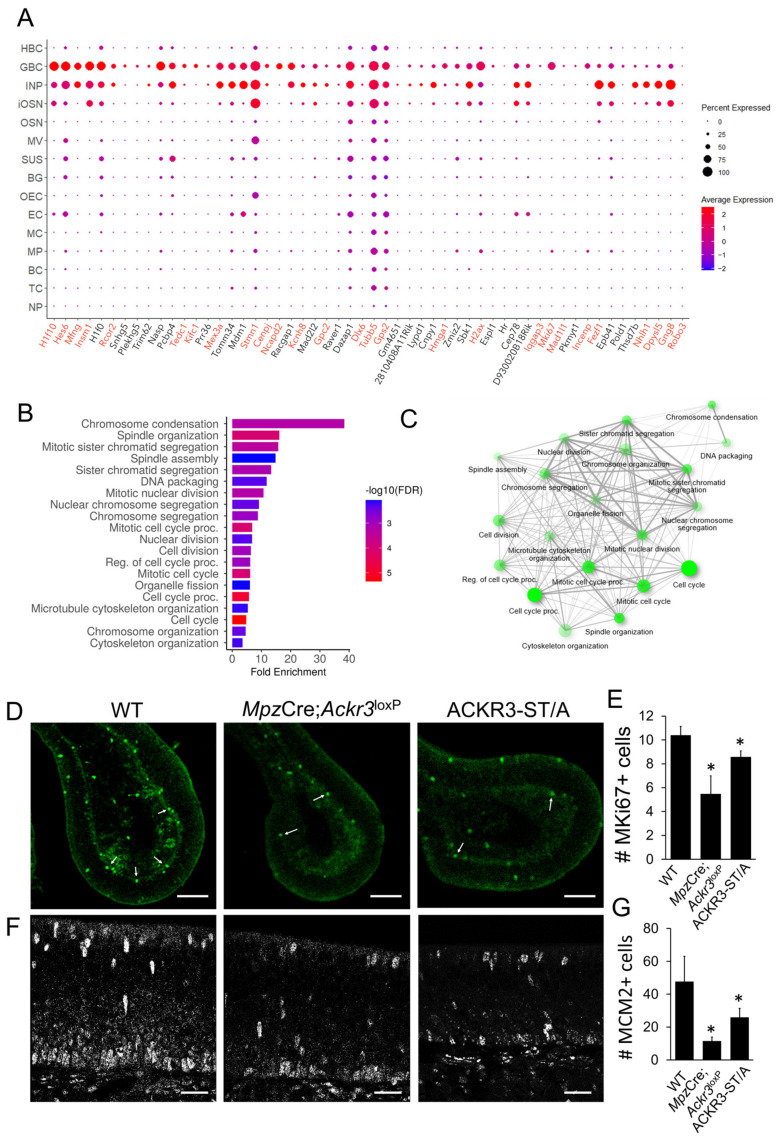
CXCL12 scavenging is required to regulate proliferation of stem cells. (**A**) Dot plot of the differentially expressed genes in GBCs and INPs of WT and *Mpz*-Cre;*Ackr3*^LoxP/LoxP^ mice (P8). For each gene depicted, the size of the circle corresponds to the percentage of cells expressing the specific gene and the color reflects the average expression level of the gene within those cells. Genes known to be involved in neurogenesis are marked in red. HBC: horizontal basal cell; GBC: globose basal cell; INP: immediate neuronal precursor; iOSN: immature olfactory sensory neuron; OSN: olfactory sensory neuron; MV: microvillar cell; SUS: sustentacular cell; BG: Bowman’s gland; OEC: olfactory ensheathing cell; EC: ependymal cell; MC: myeloid cell; MP: macrophage; BC: B cell; TC: T cell; NP: neutrophil. (**B**) GO analysis performed upon the genes whose expression in GBCs and INPs differed between WT and *Mpz*-Cre;*Ackr3*^LoxP/LoxP^ mice. Genes involved in cell proliferation represent the most differentially expressed genes. (**C**) Network of differentially expressed genes showing all pathways (nodes) are connected. Bigger nodes represent larger gene sets. Thicker edges represent more overlapped genes. (**D**) Immunolabeling of KI67 in the OE of WT, *Mpz*-Cre;*Ackr3*^LoxP/LoxP^, and ACKR3-ST/A mice. Scale bar 100 µm. (**E**) Quantification of the number of MKI67-positive cells, n minimum of 3 animals per group, Student’s *t*-test, error bars represent SEM, * *p* < 0.05. (**F**) Immunolabeling of MCM2 in the OE of WT, *Mpz*-Cre;*Ackr3*^LoxP/LoxP^, and ACKR3-ST/A mice. Scale bar 20 µm. (**G**) Quantification of the number of MKI67-positive cells, n minimum of 3 animals per group, Student’s *t*-test, error bars represent SEM, * *p* < 0.05.

**Figure 5 cells-12-02164-f005:**
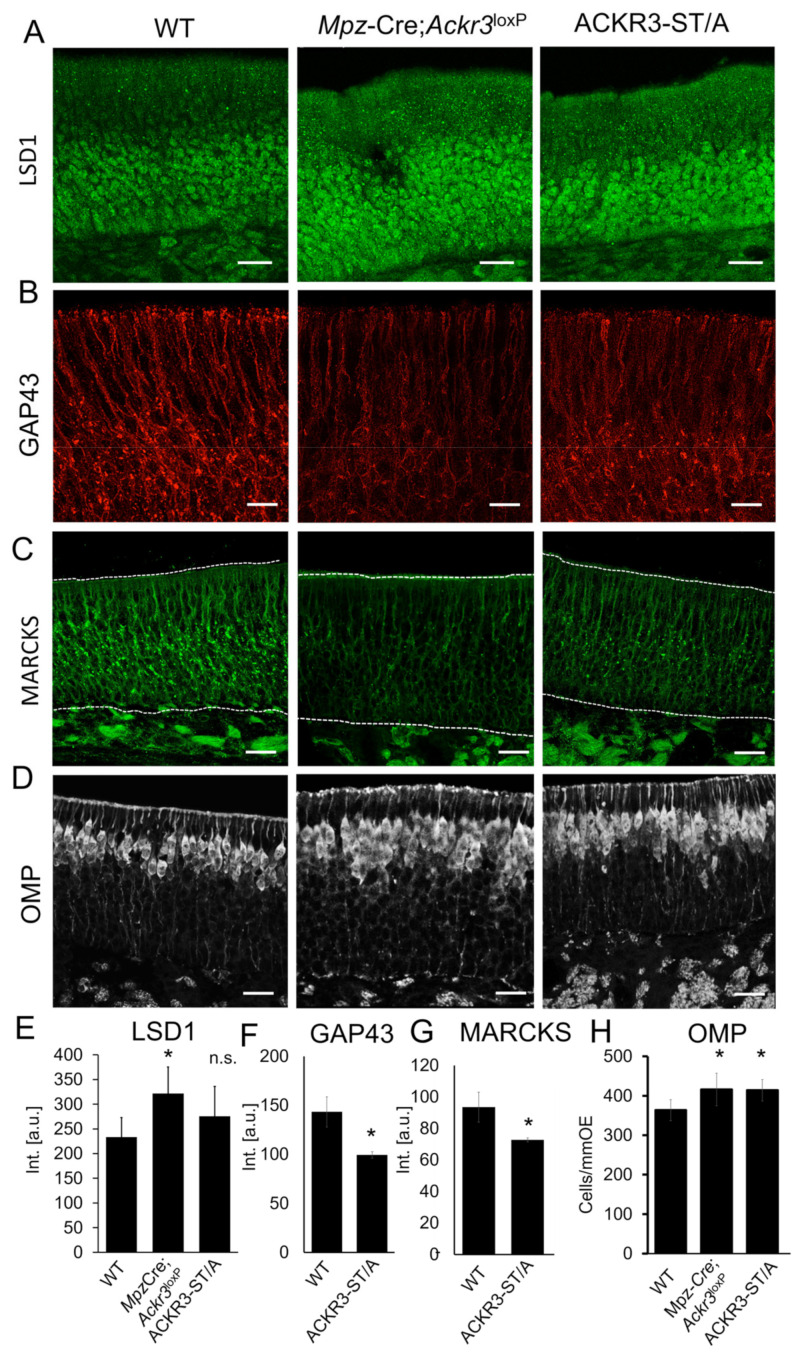
Development of olfactory neurons depends on CXCL12. Immunolabeling of different markers for neurogenesis in WT, *Mpz*-Cre;*Cxcl12*^LoxP/LoxP^, and ACKR3-ST/A mice (P8). (**A**) LSD1, (**B**) GAP43, (**C**) MARCKS, and (**D**) OMP. (**E**) Quantification of the number of LSD1-positive neurons in mice of different genotypes (P8). n.s. means “not significant”. (**F**) Quantification of the GAP43 intensities. (**G**) Quantification of the MARCKS intensities. (**H**) Quantification of the number of OMP-positive neurons. All quantifications from n minimum of 3 animals per group, Student’s *t*-test, error bars represent SEM, * *p* < 0.05. Scale bars 10 µm.

**Figure 6 cells-12-02164-f006:**
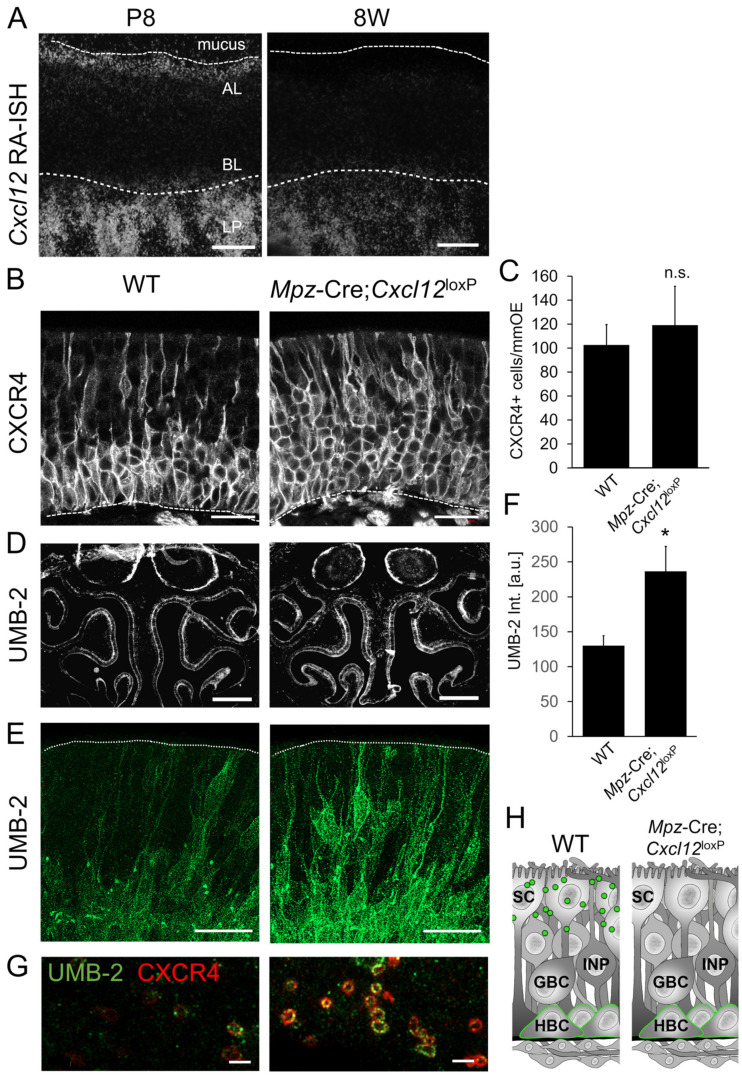
*Cxcl12* expressed in sustentacular cells activates CXCR4. (**A**) Dark field micrographs of the olfactory mucosa of P8, P28, and P56 old mice after in situ hybridization with ^35^S-labeled probes for *Cxcl12*. Highly sensitive detection reveals minor expression of *Cxcl12* in the sustentacular cell layer (in the apical layer, AL) and strong expression in the lamina propria (LP). Dotted lines demark the apical border of the OE, which is covered by mucus, and the basal lamina. CXCL12 sense probe did not show a signal (not shown). (**B**) Confocal images of coronal OE sections stained against CXCR4 protein; *Mpz*-Cre;*Cxcl12*^LoxP/LoxP^ mice showed somewhat increased membrane labeling compared to WT mice. (**C**) Quantification of CXCR4-positive cells in cryosections (n = 3 animals per group, Student’s *t*-test, error bars represent SEM, * *p* < 0.05). n.s. means “not significant”. (**D**) Low-magnification overview pictures of UMB-2 (CXCR4 non-phosphorylated) immunolabeling in WT and *Mpz*-Cre;*Cxcl12*^LoxP/LoxP^ mice (P8); shown are pictures taken with a stereomicroscope. (**E**) High-resolution Airyscan microscopy of UMB-2 (CXCR4 non-phosphorylated) immunolabeling in *Mpz*-Cre;*Cxcl12*^LoxP/LoxP^ mice (P8) showing that activation of CXCR4 in dendrites and localization of non-phosphorylated CXCR4 depends on CXCL12 availability. Shown are projections of confocal sections (16 µm); dotted lines delineate the apical surface of the OE. (**F**) Quantification of UMB-2 staining intensities in cryosections (n = 3 animals per group, Student’s *t*-test, error bars represent SEM, * *p* < 0.05). (**G**) Horizontal confocal sections through the layer of dendritic knobs. The number of knobs that are positive for CXCR4 (red) or non-phosphorylated CXCR4 (green) is increased labeling in *Mpz*-Cre;*Cxcl12*^LoxP/LoxP^ mice showing dependence on CXCL12. (**H**) Schematic overview of CXCL12 (green) localization in sustentacular cells (SC) and HBCs of the OE of WT and *Mpz*-Cre;*Cxcl12*^LoxP/LoxP^ mice. Scale bars 50 μm (**A**); 20 µm (**B**,**C**); 500 µm (**D**); 2 µm (**G**).

**Figure 7 cells-12-02164-f007:**
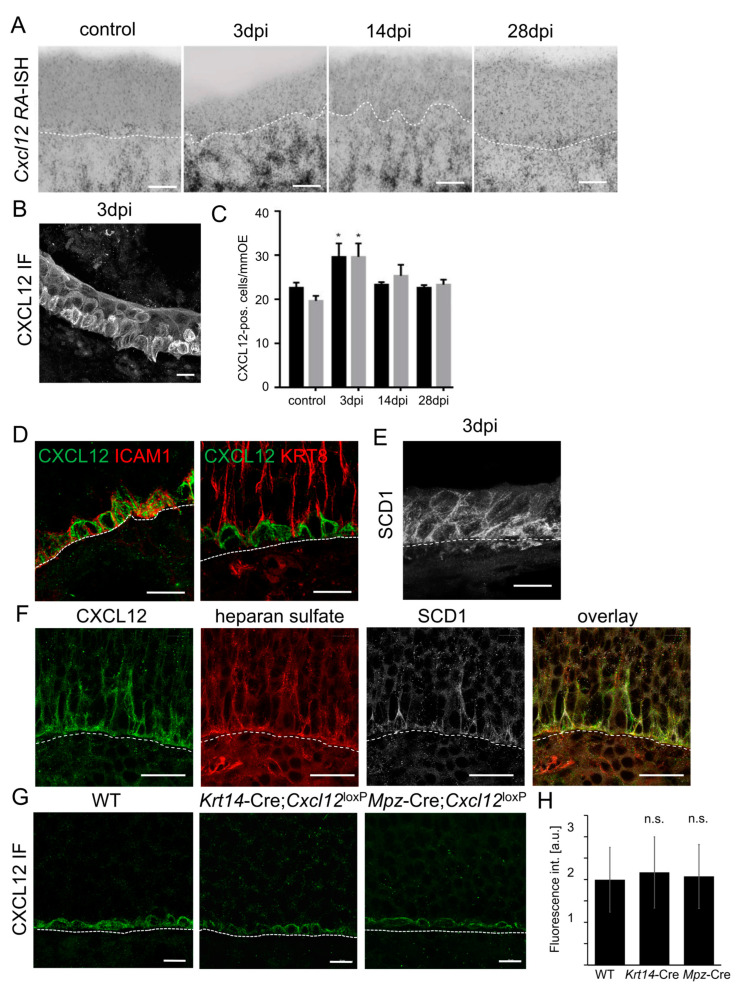
CXCL12 in HBCs is derived from the lamina propria. (**A**) Expression of *Cxcl12* during injury-induced neurogenesis. Confocal microscopy of the OE of P56 old mice (P56) after in situ hybridization with ^35^S-labeled probe for *Cxcl12*; shown are epithelia 3, 14, and 28 days after methimazole injection. *Cxcl12* mRNA expression in the LP is markedly increased at 3 days. *Cxcl12* sense probe does not show a signal. (**B**) CXCL12 immunofluorescence during regeneration (3 dpi) showing presence of the protein in the OE, which consists solely of activated HBCs at this time point. (**C**) Quantification of CXCL12-positive cells in the OE after immunostaining (n = 3 animals per group, Student’s *t*-test, error bars represent SEM, * *p* < 0.05). (**D**) Co-localization of CXCL12 (green) with ICAM1 (red) in HBCs, but no apparent co-localization of CXCL12 with the sustentacular cell marker KRT8 in the basal olfactory mucosa. (**E**) SCD1 immunofluorescence during regeneration (3 dpi) showing presence of the protein throughout the OE consisting of activated HBCs. (**F**) Co-localization of CXCL12 (green) with HS (red) and SDC1 (white) in HBCs of P8 mice. (**G**) Localization of CXCL12 in the OE from WT, *Krt14*-Cre;*Cxcl12*^LoxP/LoxP^, and *Mpz*-Cre;*Cxcl12*^LoxP/LoxP^ mice (2M). CXCL12 localization was not very different in mice of the different genotypes, showing that CXCL12 in HBCs is not synthesized in the OE, but derived from cells in the lamina propria. (**H**) Quantification of CXCL12 staining shown in (**F**). n.s. means “not significant”. Scale bars (**A**) 50 μm; (**D**,**E**) 10 μm; (**C**,**G**) 20 μm.

**Figure 8 cells-12-02164-f008:**
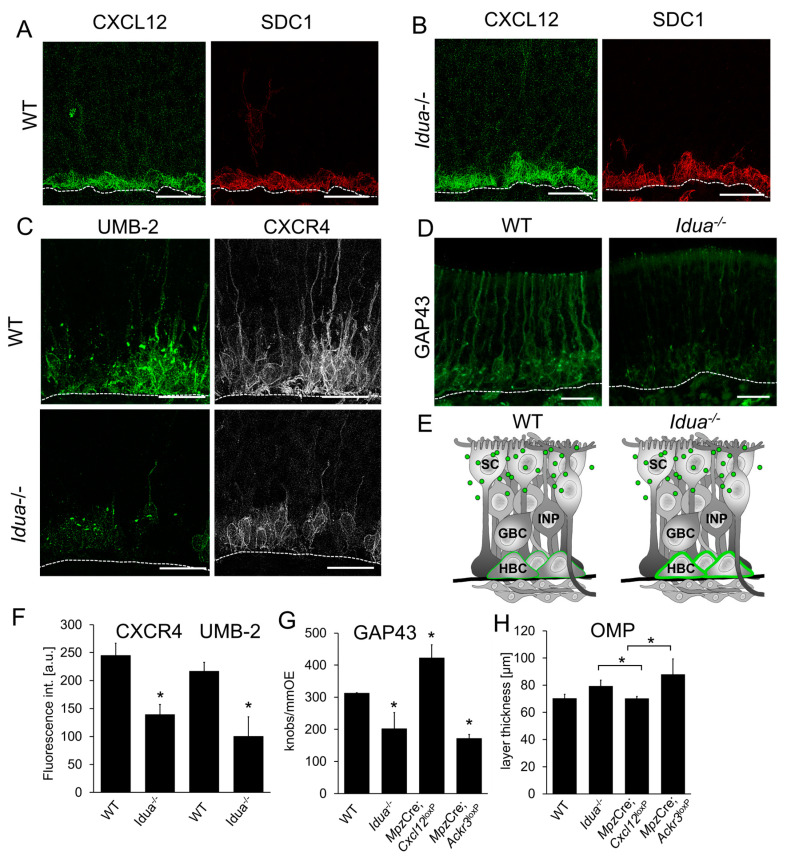
CXCL12 in HBCs is associated with HS. (**A**) Co-localization of CXCL12 (green) and SDC1 (red) in adult (8W) WT (**A**) and *Idua*^−/−^ mice (**B**). (**C**) Immunostaining for the CXCL12 receptor CXCR4 (white) and UMB-2 (non-phosphorylated CXCR4, green) in the OE of WT mice and *Idua^−/−^* mice, showing that the amount of CXCR4 on the plasma membrane is reduced. (**D**) Immunofluorescence staining showing reduced number of GAP43-positive cells in *Idua*^−/−^ mice. (**E**) Schematic overview of CXCL12 (green) localization in horizontal basal cells (HBC) of the OE of WT and *Idua*^−/−^ mice. Quantification of immunofluorescence stainings showing (**F**) reduced number of CXCR4- and UMB-2 positive cells in *Idua*^−/−^ mice, (**G**) reduced number of GAP43-positive immature neurons in *Idua*^−/−^ mice, (**H**) number of OMP-positive OSNs in *Idua*^−/−^*, Mpz*-Cre;*Ackr3*^LoxP/LoxP^, and *Mpz*-Cre;*Cxcl12*^LoxP/LoxP^ mice (n = 3 animals per group, Student’s *t*-test, error bars represent SEM, * *p* < 0.05). Scale bars (**A**,**B**): 20 μm, (**C**,**D**): 20 μm.

## Data Availability

All data are included in the manuscript.

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
