# Peer review of "Glia Cells Control Olfactory Neurogenesis by Fine-Tuning CXCL12"

_cells, 2023, doi:10.3390/cells12172164_

Round 1
Reviewer 1 Report
General comments:
This paper seems interesting because it adds a new perspective to the olfactory epithelial regeneration.
Minor comments:
1. Scales were missing in some figures, please correct them.
2. Please mention the limitation of this study in your discussion about the lack of a large number of mice studied.
Line 208-209
A single dose of 700 μl ice‐cold PBS was injected und mucus fluid was collected from the nos‐ 209 trils.
Please correct “und” to “and”.
Author Response
Please find below our specific answers:
1. Scales were missing in some figures, please correct them.
We now added scales bars to all figures. We wanted to omit the scale bars from pictures of identical magnification to improve the visibility of the stainings. We completely agree that this may be misleading and looks like missing scale bars.
2. Please mention the limitation of this study in your discussion about the lack of a large number of mice studied.
We added the requested statement to the end of the discussion (lines 733-736).
Comments on the Quality of English Language
Line 208-209
A single dose of 700 μl ice‐cold PBS was injected und mucus fluid was collected from the nos‐ 209 trils.
Please correct “und” to “and”.
We apologize for this mistake and corrected it in the revised version.
Reviewer 2 Report
Is a comprehensive work approaching the research subject from different angles.
Overall, it demonstrates that the tight regulation of CXCL12 is important for sustained neurogenesis in the olfactory stem cell niche in mice.
Line 250. “at his age”. Please fix this typo
Line 252. “Co‐hybridization of Ackr3 with a Cxcr4‐specific riboprobe showed that transcripts of both receptors do not co‐localize, ruling out that ACKR3 modulates CXCR4 by heterodimer formation”. In my opinion, the absence of co-localisation of transcripts still does not rule out the possibility of the protein heterodimer formation. The immunostaining data showing that ACKR3 and CXCR4 are localised in different cell types (sustentacular cells and immature neurons, respectively) suggests the absence of heterodimer formation.
Line 413. “[46].The”. Please fix the typo (space between the words is missing)
The partial overlap of Rcor2 and Ackr3 pathways is of interest.
The authors demonstrated that LSD1 levels are increased in Mpz‐Cre;Ackr3LoxP/LoxP mice. What about changes in its genomic loclalization? What about levels of LSD1 in Rcor2 depleted cells?
Could the authors please discuss in more detail significance of this finding and propose molecular underlying mechanisms of such overlap?
I suggest, elaborating on translational value of the findings from this work will make it even more attractive for the readers. What might be practical “take home” messages resulting from this work for whose working in translational biomedicine field?
Author Response
- Line 250. “at his age”. Please fix this typo
We apologize for this and fixed this typo.
- Line 252. “Co‐hybridization of Ackr3 with a Cxcr4‐specific riboprobe showed that transcripts of both receptors do not co‐localize, ruling out that ACKR3 modulates CXCR4 by heterodimer formation”. In my opinion, the absence of co-localisation of transcripts still does not rule out the possibility of the protein heterodimer formation. The immunostaining data showing that ACKR3 and CXCR4 are localised in different cell types (sustentacular cells and immature neurons, respectively) suggests the absence of heterodimer formation.
You are completely right, we changed the text according to your suggestion (line 258).
- Line 413. “[46].The”. Please fix the typo (space between the words is missing)
We apologize for this and fixed this typo.
- The partial overlap of Rcor2 and Ackr3 pathways is of interest.
The authors demonstrated that LSD1 levels are increased in Mpz‐Cre;Ackr3LoxP/LoxP mice. What about changes in its genomic loclalization? What about levels of LSD1 in Rcor2 depleted cells?
Could the authors please discuss in more detail significance of this finding and propose molecular underlying mechanisms of such overlap?
We now specifically mention LSD1/Rcor2 activation in the discussion (lines 714-723). The effect of ACKR3 on LSD1/RCOR2 is indirect. ACKR3 is present in sustentacular cells, while LSD1 increase is observed in the neuronal stem and progenitor cells. ACKR3 has an impact since it alters the CXCL12 concentration and thereby regulates CXCR4 signaling. We have not yet investigated the signaling cascade downstream of CXCR4 in the stem cells, but CXCR4 is well known to trigger the phosphatidylinositol 3-kinase (PI3K) pathway, leading to the phosphorylation and activation of Akt, which could then influence gene expression in the nuclear compartment. However, also other signaling cascades have been described and could impact the activation. The details of the signaling cascade of CXCR4 in olfactory stem cells would have to await further experimentation. It would for sure be interesting to investigate the LSD1 levels in Rcor2 depleted cells, but we cannot produce these cells/the transgenic animals in short time. We also did not investigate the genomic localization, which would indeed be an interesting follow up of this study.
- I suggest, elaborating on translational value of the findings from this work will make it even more attractive for the readers. What might be practical “take home” messages resulting from this work for whose working in translational biomedicine field?
We thank the reviewer for his / her helpful comment. We added a statement with regard to the potential translation implications to the end of the conclusion (lines 749-754), which are for sure very speculative.
Reviewer 3 Report
This paper investigates the role of CXCL122 in the olfactory epithelium with focus on regulation of its receptor and ECM. This is a well written careful study with lovely IHC images. Since cxcl12 can affect immune cells and neural cells inclusion of immune marker and an olfactoyr glia marker could have been useful and informative.
Specific points.
1. CXCL12 is known also tor regulate myelination and maybe this could be discussed. I acknowledge that the olfactory receptor neurons are not myelinated.
2. IS PO expressed in the olfactory ensheathing cells? Could this affect work. Maybe some discussion on these cells could have been made?
3. A table of the transgenic mice and phenotype would have been extremely useful.
4. When SDFα is used it should be explained, is this the same as SDF1-alpha which is CXCL12? (page 5)
5. Some more description of PLA methodology would have been helpful in the methods.
6. Since Olfactory glia can secrete cxcl12, some consideration for these cells should have been made. (Jiang C, Wang X, Jiang Y, Chen Z, Zhang Y, Hao D, Yang H. The Anti-inflammation Property of Olfactory Ensheathing Cells in Neural Regeneration After Spinal Cord Injury.)
7. CXCL12 can affect immune cells and maybe olfactory glia, so inclusion of these markers would complement this work, eg P75, microglia or T cell marker ?
Author Response
- CXCL12 is known also tor regulate myelination and maybe this could be discussed. I acknowledge that the olfactory receptor neurons are not myelinated.
Thank you for this valuable comment. We agree on your point and added a statement with regard to myelination in the discussion. Since, as already mentioned by yourself, olfactory axons are not myelinated and we did not observe a difference in the overall structure of the axon bundles, the statement is rather short. We combined this point and adressed your points 2 and 6 in the same paragraph (lines 663-675 in the discussion, highlighted yellow). We now mention that CXCL12 could likely be secreted from from olfactory ensheathing cells and also from other cells of the laminar propria, since Mpz-Cre;Cxcl12loxp/loxp mice still express Cxcl12 in the lamina propria. We also mention that MPZ is expressed in the lamina propria, most likely in olfactory ensheathing cells. We have no data showing that whether all ensheathing cells express MPZ, since it is well known that different subpopulations exist. Moreover, ensheathing cells may be exist in different levels of activation. The identification of the contribution of the different cell types of the lamina propria expressing CXCL12 would clearly be beyond the scope of this manuscript.
- IS PO expressed in the olfactory ensheathing cells? Could this affect work. Maybe some discussion on these cells could have been made?
Please see our comment to point 1, where we adressed this point.
- A table of the transgenic mice and phenotype would have been extremely useful.
We added a table summarizing the different genotypes (Supplementary table 3 in the revised manuscript).
- When SDFα is used it should be explained, is this the same as SDF1-alpha which is CXCL12? (page 5)
We apologize for this ambiguity. CXCL12 was formerly called SDF1, with SDFalpha being the most abundant isoform. Our in-situ probes are directed against this isoform. The commercial kit used to detect CXCL12 is called SDF Elisa by the manufacturer. We now added CXCL12 in brackets to make this point clear in the Methods part.
- Some more description of PLA methodology would have been helpful in the methods.
We extended the description of the PLA methodology according to your suggestions (lines 137-155).
- Since Olfactory glia can secrete cxcl12, some consideration for these cells should have been made. (Jiang C, Wang X, Jiang Y, Chen Z, Zhang Y, Hao D, Yang H. The Anti-inflammation Property of Olfactory Ensheathing Cells in Neural Regeneration After Spinal Cord Injury.)
Please see our comment to point 1, where we adressed this point. We also cited the suggested paper in this context.
- CXCL12 can affect immune cells and maybe olfactory glia, so inclusion of these markers would complement this work, eg P75, microglia or T cell marker ?
We did not analyse the subpopulations or activation state of olfactory ensheathing cell in the context of this manuscript. We focussed on the stem cell niche, which already constitues a quite complex set of genotypes and experiments in our view. An investigation on the role of ensheathing cells and immune cells in the context of potentially altered CXCL12 levels is for sure interesting. Actually we observed a phenotype more complex than just CXCL12 alteration in Mpz-Cre;Ackr3loxp/loxp mice and we are currently investigating the composition of the lamina propria in detail.
Round 2
Reviewer 3 Report
The paper has been adjusted appropriately/.